# Prevalence of ADHD in Accident Victims: Results of the PRADA Study

**DOI:** 10.3390/jcm8101643

**Published:** 2019-10-08

**Authors:** Sarah Kittel-Schneider, Sarah Wolff, Kristin Queiser, Leonie Wessendorf, Anna Maria Meier, Moritz Verdenhalven, Nathalie Brunkhorst-Kanaan, Oliver Grimm, Rhiannon McNeill, Sascha Grabow, Christoph Reimertz, Christoph Nau, Michelle Klos, Andreas Reif

**Affiliations:** 1Department of Psychiatry, Psychosomatic Medicine and Psychotherapy, University Hospital Frankfurt, Goethe-University, D-60528 Frankfurt/Main, Germany; srhwolff@yahoo.de (S.W.); mail.to.kristin@gmx.de (K.Q.); leonie-wessendorf@t-online.de (L.W.); annamariameier92@icloud.com (A.M.M.); moritz.verdenhalven@kgu.de (M.V.); nathalie.brunkhorst-kanaan@kgu.de (N.B.-K.); oliver.grimm@kgu.de (O.G.); sascha.grabow@kgu.de (S.G.); andreas.reif@kgu.de (A.R.); 2Department of Psychiatry, Psychosomatic Medicine and Psychotherapy, University Hospital Würzburg, Julius-Maximilians-University Würzburg, D-97080 Würzburg, Germany; mcneill_r@ukw.de; 3Berufsgenossenschaftliche Unfallklinik, D-60389 Frankfurt/Main, Germany; christoph.reimertz@bgu-frankfurt.de; 4Department Trauma, Hand and Reconstructive Surgery, University Hospital Frankfurt, Goethe-University, D-60596 Frankfurt/Main, Germany; christoph.nau@kgu.de; 5Department of Oral and Maxillofacial Surgery, University Hospital Frankfurt, Goethe-University, D-60596 Frankfurt/Main, Germany; michelle.klos@kgu.de

**Keywords:** adult attention deficit/hyperactivity disorder (adult ADHD), accidents, psychosocial stress, cross-sectional study

## Abstract

Background: Recent research has shown an increased risk of accidents and injuries in ADHD patients, which could potentially be reduced by stimulant treatment. Therefore, the first aim of our study was to evaluate the prevalence of adult ADHD in a trauma surgery population. The second aim was to investigate accident mechanisms and circumstances which could be specific to ADHD patients, in comparison to the general population. Methods: We screened 905 accident victims for ADHD using the ASRS 18-item self-report questionnaire. The basic demographic data and circumstances of the accidents were also assessed. Results: Prevalence of adult ADHD was found to be 6.18% in our trauma surgery patient sample. ADHD accident victims reported significantly higher rates of distraction, stress and overconfidence in comparison to non-ADHD accident victims. Overconfidence and being in thoughts as causal mechanisms for the accidents remained significantly higher in ADHD patients after correction for multiple comparison. ADHD patients additionally reported a history of multiple accidents. Conclusion: The majority of ADHD patients in our sample had not previously been diagnosed and were therefore not receiving treatment. The results subsequently suggest that general ADHD screening in trauma surgery patients may be useful in preventing further accidents in ADHD patients. Furthermore, psychoeducation regarding specific causal accident mechanisms could be implemented in ADHD therapy to decrease accident incidence rate.

## 1. Introduction

Attention-deficit/hyperactivity disorder (ADHD) is among the most prevalent neurodevelopmental disorders worldwide, with a prevalence of ~5% in children and ~2%–3% in adults [1,2,3]. The disorder is characterized by the main symptoms of inattention, hyperactivity and impulsivity, however additional symptoms can include emotional dysregulation, disorganization, sleeping problems and severe impairment in task structuring [4]. Not only do ADHD symptoms lead to impaired psychosocial functioning, but the patients are also at an increased risk of developing psychiatric comorbidities such as conduct, affective, anxiety, personality and substance abuse disorders [5,6]. In recent years an increasing amount of studies have additionally shown that ADHD is associated with a higher burden of somatic disorders, such as obesity, diabetes mellitus, asthma and migraines [7]. 

It has also been reported that ADHD patients have a higher mortality risk before the age of 40, which is mainly due to accidents [8]. Several studies and meta-analyses have consistently shown that children, adolescents, and adults with ADHD have an increased overall ~1.2 to 2-fold risk of non-intentional injuries [9,10,11,12,13]. Research investigating the specific factors that contribute to this increased injury risk in ADHD patients is important, with the aim of accident prevention. However, there are currently few studies investigating accident mechanisms in ADHD accident victims compared to non-ADHD. One study determining the cause of driving accidents found that several risk factors were increased in ADHD drivers compared to healthy controls, including speeding, risky driving, driving without a license, driving-related anger/aggression, and driving under influence of alcohol or illegal substances [14]. Furthermore, ADHD drivers were more likely to report daytime sleepiness or external distraction shortly before an accident [15,16]. Treatment with stimulant medication has been observed to reduce the risk of injuries, especially from motor vehicle accidents [17,18]. However, it is currently unclear whether non-pharmacological treatment may also be effective in ADHD patient accident prevention. 

The aims of our PRADA study (“Prevalence of ADHD in accident victims”) were to firstly investigate whether ADHD prevalence was higher in a sample of accident victims than the estimated prevalence in the general population. To achieve this, we screened a sample of accident victims for ADHD symptoms using the Adult ADHD Self-Report Scale Version 1.1 (ASRS v1.1) screening questionnaire. Given that adult ADHD at least in Germany is still severely underdiagnosed and under-treated [19,20], in the future it might be meaningful to establish a general screening for ADHD in trauma surgery units. Our second aim was to assess whether there were different accident mechanisms between ADHD accident victims and non-ADHD accident victims, in order to determine possible non-pharmacological accident prevention strategies.

## 2. Experimental Section

### 2.1. Participants

The sample cohort was recruited in three trauma surgery units; the Occupational Accident Clinic (Berufsgenossenschaftliche Unfallklinik), Frankfurt; the Department of Trauma, Hand and Reconstructive Surgery, University Hospital Frankfurt; and the Department of Oral, Dental and Cosmetic Facial Surgery, University Hospital Frankfurt. All units are based in Frankfurt, Germany. Participants were recruited between November 2016 and May 2018 within the framework of the “Prevalence of Adult Attention Deficit/Hyperactivity Disorder in Accident Victims” (PRADA) study. A preliminary analysis of a subsample of this study has already been published [21]. During the study period, a total of 3175 patients after accidents or injuries (ICD-10 diagnoses S00-T98) were treated in the two trauma surgery units at the University Hospital Frankfurt, and 3234 patients were treated in the Occupational Accident Clinic. From the total 6409 trauma surgery patients we were able to recruit 905 accident victims, and those were screened for ADHD symptoms using the ASRSv1.1 18 item self-report questionnaire as well as the ASRS short form (ASRS-SF) [22,23]. Information regarding age and sex was collected from all participants. 

Participants that screened positive for ADHD were asked for further information regarding the circumstances of the accident that led to admittance in the trauma unit, using a self-designed questionnaire (for the accident questionnaire translated into English please see Appendix A). Accident circumstances of participants who screened positive for ADHD were then compared to a randomly picked non-ADHD subgroup of 214 trauma surgery patients, who screened negative in the ASRSv1.1. For an overview of the whole sample and subgroups see Figure 1. Participants who screened positive for ADHD were also invited to visit the specialized ADHD outpatient clinic of the Department of Psychiatry, Psychotherapy and Psychosomatic Medicine, University Hospital Frankfurt, for a deeper diagnostic assessment and potential treatment. In addition to admission to the trauma surgery department post-accident, the inclusion criteria consisted of a minimum age of 18 years and sufficient German language skills to understand the instruments used in our study. Lastly, only study participants who gave written informed consent were enrolled in the study, which complied with the latest Declaration of Helsinki and was approved by the Ethics Committee of the University Hospital of Frankfurt.

### 2.2. Questionnaires

#### 2.2.1. Adult ADHD Self-Report Scale Version 1.1 (ASRSv1.1) 

We used the World Health Organization’s ASRS v1.1 18 item questionnaires for ADHD screening [22]. The ASRS-18 is a valid instrument for ADHD screening and diagnosis with a sensitivity of 56.3%, specificity of 98.3%, and a total classification accuracy of 96.2% and kappa 0.58 [24]. In our study the German version of the ASRS v1.1 was used. Reliability and validity of the German ASRS v1.1 were previously tested by Buchli-Kammermann et al. with a sensitivity of 66.6%, specificity of 64.9%, and positive predictive value of 0.84 [23]. The ASRSv1.1 is an 18-item checklist that assesses symptoms of ADHD in adult populations. The ASRS contains two components, Part A and Part B. Part A consists of six items; four regarding attention deficit symptoms and two regarding hyperactivity/impulsivity symptoms. Part A of the ASRS has shown equal sensitivity to the full version and is commonly used as a separate screening tool, known as the ASRS v1.1 short form (ASRS-SF). Participants completed the full-version of the ASRS (ASRS-18) indicating the frequency of symptom occurrence 0 (never), 1 (rarely), 2 (sometimes), 3 (often) or 4 (very often). The ASRS-SF can be considered positive when four or more answers are above the cut-off value, which equates to four or more questions answered with ≥2 (questions 1–3) and/or ≥3 (questions 4–6) [24]. Part B consists of 12 additional ADHD symptom assessments. In the current study, patients were only included in further analyses following a positive screen on the ASRS-SF. Those who did not screen positive in the ASRS-SF were not included in primary analyses [25]. Post-hoc analyses were run on the full ASRS-18, regardless of positive screens, to determine any ADHD sub-type influence.

#### 2.2.2. Accident Questionnaire

A semi-structured/qualitative accident questionnaire was designed for this study (see Appendix A and Appendix A). It was conducted as interviews, and included questions regarding the mechanism, type, and place of the participants’ accidents. It also posed questions regarding potential causes of the accident, such as substance use, distraction, pre-accident stress, pre-accident emotional events, stress at the time of recruitment (current stress), overconfidence and fatigue/lack of sleep. Furthermore, we asked for patient medical and pharmaceutical history, and occurrence of other accidents in the past year. All data were based on self-reporting.

### 2.3. Statistical Analysis

Data analysis was conducted using SPSS (Statistical Package for Social Science), Version V24 (IBM^®^ SPSS^®^ Statistics V24, Armonk, USA). We compared a randomly selected subgroup of the non-ADHD accident victims (from herein called the control group; *n* = 214) and accident victims who screened positive for ADHD in the ASRS short form (from herein called the ADHD group; *n* = 56) with regards to sex and age distribution, and all items of the accident questionnaire, by Chi-square Test. If case numbers were too small, the Fisher’s Exact Test was utilized. The student’s t-test was used to compare the age of the tested groups. As the accident questionnaire consisted of 22 different items, we applied Bonferroni-correction for multiple testing, and all p-values reported were defined as being significant at the level of *p* ≤ 0.0023 (0.05/22). Due to the age difference between the ADHD group and the control group, all items which showed significant results in the Chi-square Test/Fisher’s Exact Test were later included in an additional multivariate logistic regression model.

## 3. Results

### 3.1. ADHD Prevalence

In the total accident victim sample of 905 participants, 56 participants screened positive for adult ADHD in the ASRS-SF. 19 additional participants screened negative in ASRS-SF (only 3 items above the cutoff instead of 4), but scored highly in Part B of the ASRS-18. Using only the positively screened participants in the ASRS-SF, there was a prevalence of 6.18 % adult ADHD in our sample of accident victims. A less conservative approach would have demonstrated a prevalence of 8.3%, including the additional participants who screened positively for ADHD in the ASRS-18. For further analyses, we used only the group of patients who screened positive for ADHD in the ASRS-SF. From herein they are referred to as the ADHD group, and the control group consists of accident victims who screened negative in the ASRS short version. 

From the 75 patients that screened positively for ADHD using both the short and full versions of the ARSR, only 14.7% (*n* = 11) visited our specialized outpatient clinic to get a full diagnostic assessment. From these participants, seven were confirmed to have adult ADHD (persistent; 63.6%), two had confirmed childhood ADHD but did not fulfill the full adulthood criteria (partially persistent; 18.2%), and two had the initial ADHD diagnosis dismissed (18.2%). One of these dismissals was due to the presence of multiple substance abuse disorder, and one due to a previous head trauma, which could have caused the cognitive and behavioral ADHD-like symptoms. Combining participants with persistent and partially persistent ADHD, the positive ASRS screening could be validated in 81.8 % of participants. Adjusting our initial conservative ADHD prevalence of 6.18% based on the assumption that ~80% diagnoses would be confirmed as either persistent or partially persistent ADHD in a full assessment, consequently resulted in a corrected estimated prevalence of 4.88% adult ADHD in our accident victim sample. This ADHD prevalence is approximately 1.6-fold higher than in the general population. 

### 3.2. Differences in Demographic Data Between ADHD and Non-ADHD Accident Victims

There were more male patients (65.3% vs. 34.7%) than female patients in the total accident victim cohort. In the ADHD group (following the more conservative screening method) there were 28.6% females and 35.3% females in the control group, which was not statistically significant. However, ADHD patients were found to be significantly younger than the control trauma patients (*p* < 0.0001, see Table 1).

More detailed demographic data regarding comorbidities, medication, and accident information were only collected from the ADHD group and the control group of 214 randomly selected non-ADHD accident victims. Within this smaller ADHD and control subgroup, ADHD patients were still significantly younger, whereas sex differences were similar (28.6% female in the ADHD group vs. 31.3% females in the selected control group). These subgroup findings were comparable with the total accident victim cohort findings. 

Participants from the ADHD group had a significantly higher number of psychiatric comorbidities such as anxiety disorders, adjustment disorder, schizophrenia, substance abuse and chronic pain condition. However, only substance abuse remained statistically significant after correction for multiple comparisons (χ²-test, *p* < 0.0001). A pre-existing, previously confirmed ADHD diagnosis was also significantly more common in our ADHD group (χ²-test = 21.50, *p* < 0.0001) compared to controls, further validating the screening instrument used. However, there were two participants in the control sample who screened negative for ADHD in the ASRS, yet reported having an ADHD diagnosis. Lastly, ADHD patients were significantly more likely to be regularly taking prescribed sedatives and other psychopharmacological medication (χ²-test, *p* < 0.0001) (see Table 2).

### 3.3. Circumstances and Mechanisms of Accidents 

In the total sample, most accidents happened in road traffic, but after calculating a logistic regression there were no significant differences between ADHD patients and controls regarding the place of accidents or the type of road accident (pedestrians, motorbike, care, bicycle etc.). Self-inflicted accidents were also not significantly higher in the ADHD group based on self-report (χ²-test =104.32, *p* = 0.06). However, ADHD patients were significantly more likely to be under the influence of alcohol or illegal drugs when the accident happened (χ²-test, *p* all *p* < 0.0001). Medication use directly before the accident did not remain statistically significant different after correcting the χ²-test for multiple testing, and no medication was significantly different between ADHD and non-ADHD (all *p* ≤ 0.038). ADHD patients were significantly more likely to have been generally distracted before the accident (χ²-test = 6.21, *p* = 0.01; logistic regression, *p* = 0.009), however this finding was no longer statistically significant after correcting for multiple comparisons. Mind wandering/being in thoughts before the accident was statistically higher in the ADHD group (χ²-test = 9.41, *p* = 0.002). 

During recruitment and interview, significantly more ADHD patients reported feeling stressed, however this did not withstand multiple corrections (χ²-test = 8.76, *p* = 0.003). Despite this, significantly more ADHD patients reported feeling stressed at the time the accident happened (χ²-test = 9.75, *p* = 0.002). This result was already reported in our initial study, and has now been validated in the total accident victim cohort, using a greater number of control patients [21]. The most commons reasons given for being stressed, particularly in the ADHD group, were having an argument or being under pressure. Furthermore, the reported accident cause of being overconfident was significantly higher in the ADHD group (χ²-test = 15.33, *p* = 0.0004). Lack of sleep or tiredness was not statistically significant different between the groups (χ²-test = 5.67, *p* = 0.058). 

Additionally, multivariate logistic regression analysis including diagnosis and variables that were identified as significantly different between the ADHD and control groups in the single tests (age, substance influence in general, alcohol, illegal drugs, being in-thoughts, being stressed before the accident, having had an argument and overconfidence) revealed that only being in thoughts and being overconfident remained significantly different between ADHD and controls (see Table 3). Lastly, the ADHD group reported a significantly higher number of previous accidents compared to controls (χ²-test = 16.71, *p* < 0.0001) (see Table 4).

## 4. Discussion

In recent years, it has been consistently reported that childhood, adolescent and adult ADHD is associated with an increased risk of unintentional injuries and accidents. A recent meta-analysis (including data from 20,363/350,938 ADHD cases vs. 901,891/4,055,620 non-ADHD HD controls for Hazard Ratio/Odd’s Ratio analysis) concluded that the risk of injuries in ADHD is increased by 1.53 OR and 1.38 HR, respectively [12]. Dalsgaard and colleagues found an increased mortality rate in ADHD which was mainly due to accidents [8]. Furthermore, there is evidence that ADHD patients additionally are at risk for multiple trauma and multiple accidents [25,26]. Thus, we hypothesized that in population of accident victims a higher number of ADHD patients compared to general population should be found. As well, if this assumption should be correct, because of the risk of subsequent accidents, a screening for ADHD might be necessary in trauma patient population. Sufficient medication with stimulants has already been shown to decrease the accident risk, so it would be important to diagnose and identify the ADHD patients and enable them being adequately treated [27]. 

We performed a cross-sectional study using a cohort of 905 accident victims recruited from three separate trauma surgery units, and screened for adult ADHD using the ASRS-18 item questionnaire as well as the ASRS short form [28]. We found an adult ADHD incidence rate of 6.18%, which is markedly higher than the prevalence estimates for the general worldwide and German populations, which range from 3.4% to 4.7% [29,30]. When using a less conservative version of the ASRS-SF, the ASRS-18, to assess participants, the estimated prevalence of adult ADHD increased to 8.3% in our cohort [31]. Our results strengthen previous findings of increased risk of injuries and accidents in adult ADHD patients. A recent matched-cohort study from Taiwan reported an increased risk of 143% for overall injuries in adult ADHD patients compared to non-ADHD controls [32]. Using insurance data, accident risk in children, adolescent and adult ADHD patients was additionally estimated to be 1.7 fold higher than in non-ADHD controls [33]. Additionally, ADHD patients in our sample reported having had more or multiple accidents in the past compared to non-ADHD patients, again consistent with previous findings [25,26]. 

Demographic data showed that ADHD patients had a higher number of psychiatric comorbidities, particularly substance use disorders and chronic pain, consistent with previous findings [34,35]. ADHD patients were also more likely to be taking prescribed sedatives and other psychopharmacological medication (such as antidepressants and antipsychotics). In contrast to other studies [9,36], we did not find a significant sex difference between ADHD and non-ADHD accident victims. In general, men were found to be more prone to accidents. We expected that the percentage of women in the ADHD group would be higher than in the non-ADHD group which was not the case. Our hypothesis was that women with ADHD might be much higher at risk for accidents than non-ADHD women but ADHD men might not be at a higher risk compared to ADHD women. However, this was not true in our results. This discrepancy may be due to the relatively small numbers used in our study compared to previous studies. The ADHD patients in our sample were significantly younger than the control accident victims. The reasons for this result are not clear, and therefore we can only speculate. It could be that because ADHD symptoms ameliorate and change over the life span, younger ADHD patients are more at risk for accidents. Future studies comparing accident rate in age-matched samples may be able to clarify this. Limitations of our study are that we did not assess injury type or time between trauma and assessment as well as no detailed medical record data. However, we also did not recruit patients from a neurosurgery unit, to ensure there were no patients with severe head trauma, which may have biased ADHD symptom results. 

The mechanisms underlying this increased risk of injuries and accidents in ADHD are currently unknown. Knowledge of such mechanisms could potentially lead to non-pharmacological accident prevention strategies. It has previously been suggested that ADHD patients might drive in a more-risky manner, and also more frequently under substance influence, contributing to car accidents [14]. We could confirm that ADHD patients were more often under the influence of alcohol or illegal drugs upon accident occurrence, compared to the non-ADHD trauma patients. ADHD patients have also been shown to possess slower and more variable reaction times, in addition to increase feelings of fatigue, in a driving simulation study [14]. In support of this, daytime sleepiness has been suggested as a specific risk factor for ADHD drivers [15]. In contrast, our ADHD patients did not report significantly higher feelings of fatigue or tiredness before the accident. The previous study however included 36,140 drivers, of which 1543 positively screened for ADHD, and consequently had a much larger sample size than our study. The authors also only investigated traffic accidents [15]. Therefore, daytime sleepiness may be an accident risk factor, although specifically for car accidents. Distraction may also play a more significant role in car accidents for ADHD drivers than for controls [16,37]. In our study, a significantly higher number of car accidents due to distraction (mind wandering and own thoughts) was reported in ADHD patients. Distraction may also play a role in other types of accidents, functioning as general risk factor, although further studies will be needed to clarify this. 

In our cohort, ADHD participants more frequently attributed overconfidence to accident causality. Additionally, our results supported the data obtained from our preliminary study [21], whereby ADHD patients were under higher self-perceived stress prior to the accident. In supporting of this, it has repeatedly been observed that ADHD patients experience a higher level of psychosocial stress, and display a dysregulated hypothalamus-pituitary-adrenocortical-axis (HPA axis) [38,39,40]. Moreover, it has been suggested that these factors may contribute to the high risk of developing comorbid depression in adult ADHD [41,42,43]. Additionally, stress-induced cognitive deficits may be more pronounced in ADHD patients, contributing to the increased risk of injuries and accidents. However, this result did not remain significantly different after correction for multiple comparison, so the contribution of stress to causing accidents in ADHD might be less than mind wandering and over-confidence. 

From our results, we conclude that a screening for ADHD in trauma surgery patients should be recommended, as patients may be at increased risk for multiple accidents and adult ADHD. Moreover, adult ADHD is still underdiagnosed and undertreated in Germany, and routine trauma surgery screening may help to improve this. We additionally suggest that ADHD psychoeducation should include information about accident risk, focusing on prevention of substance abuse, avoidance of overconfidence, strategies against mind-wandering and stress reduction management, as a potential non-pharmacological strategy to decrease accident incidence in ADHD patients. 

## Figures and Tables

**Figure 1 jcm-08-01643-f001:**
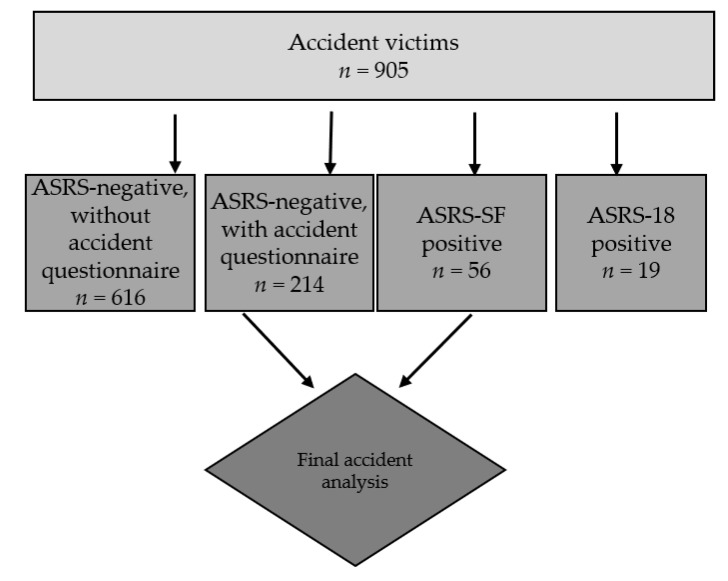
Accident victim sample cohort.

**Table 1 jcm-08-01643-t001:** Basic demographic data from total accident victim cohort.

	ADHD (ASRS-SF)	ADHD (ASRS-18)	All Controls	Selected Controls	*p*
**Sex (m/f)**	40/16	14/5	537/293	147/67	n.s.
**Age (years, mean ± SD)**	40.7 ± 15.2	39.1 ± 13.6	47.4 ± 15.3	48.6 ± 13.2	**<0.0001**

Differences in age between ADHD and control groups were calculated by student’s test/ANOVA and results given as mean ± standard deviation (SD). Differences in sex were calculated by χ²-test. Corrected level of significance was set at *p* = 0.0023. Significant *p*-values are in bold. ADHD = attention-deficit/hyperactivity disorder; m = male; f = female.

**Table 2 jcm-08-01643-t002:** Detailed demographic data from the randomly selected subgroup and the ADHD group.

	ADHD	Controls	*p*
**Sex (m/f)**	40/16	143/71	n.s.
**Age (mean ± SD)**	40.7 ± 15.2	48.6 ± 13.2	**0.001**
**Comorbidities**	*n*	*n*	
Affective Disorder	4/44	6/208	0.071
Personality Disorders	0/48	0/214	n.s.
Eating Disorder	0/48	0/214	n.s.
Anxiety Disorder	1/47	0/214	0.034
Adjustment Disorder	1/48	0/214	0.036
Schizophrenia	1/47	0/214	0.034
Substance Abuse	4/45	1/212	**<0.0001**
ADHD	7/42	2/212	**<0.0001**
Skeletal Disease	7/43	14/200	0.079
Cardiovascular Disease	9/41	54/160	n.s.
Thyroid Disease	3/46	16/198	n.s.
Metabolic Disease	6/43	13/201	n.s.
Pulmonary Disease	6/44	13/203	n.s.
Neurological Disease	5/44	10/203	n.s.
Chronic pain	5/45	3/210	**0.001**
Other	9/41	36/178	n.s.
**Regular medication (yes/no)**	26/24	96/118	n.s.
Sedatives (yes/no)	4/45	1/211	**<0.0001**
Other psychopharmacological medication (yes/no)	8/41	5/208	**<0.0001**
CVD Medication (yes/no)	8/41	57/156	n.s.
Anticoagulants/thrombocyte-aggregation-inhibitors (yes/no)	0/48	6/207	n.s.
Pain medication (yes/no)	6/42	10/203	0.042
Stimulant medication (yes/no)	2/46	0/213	0.003
Other (yes/no)	14/34	51/161	n.s.

Differences in age between ADHD and control groups were calculated by student’s test and results given as mean ± standard deviation (SD). Differences in sex, somatic and psychiatric comorbidities, and regular medication were calculated by χ²-test. Level of significance was set at *p* < 0.0023. Significant *p*-values are in bold. ADHD = attention-deficit/hyperactivity disorder; m = male; f = female; CVD = cardiovascular.

**Table 3 jcm-08-01643-t003:** Different accident circumstances between ADHD and controls.

	*p*	95% Confidence Interval	
		Lower Bound	Upper Bound
**In thoughts**	**0.04**	0.05	0.97
**Substance influence**	0.41	0.004	9.67
**Alcohol**	0.85	0.03	56.98
**Illegal drugs**	0.72	0.04	8.77
**Stressed before accident**	0.21	0.27	1.33
**Having had an argument**	0.09	0.05	1.25
**Overconfidence**	**0.02**	0.18	0.87

Multivariate Logistic regression analysis including diagnosis and variables that were identified as significantly different between the ADHD and control groups in the single tests was conducted. Level of significance was set at *p* ≤ 0.05. Significant *p*-values are in bold.

**Table 4 jcm-08-01643-t004:** Accident circumstances.

	ADHD (*n*)	Controls (*n*)	*p*
Road traffic	20	63	n.s.
At home	7	19	n.s.
Sports	8	38	n.s.
Work/School/University	10	63	n.s.
Public place	5	28	n.s.
Other	0	3	n.s.
Missing	6	0	
**Self-inflicted (yes/no)**	35/14	121/92	0.06
Missing	6	0	
**Substance influence (yes/no)**	11/39	11/202	**<0.0001**
Alcohol (yes/no)	10/40	10/203	**<0.0001**
Illegal drugs (yes/no)	5/45	2/211	**<0.0001**
Missing	6	1	
**Medication (yes/no)**	4/46	2/212	0.003
Missing	6	0	
Sedatives (yes/no)	3/47	1/212	0.004
Other psychopharmacological medication (yes/no)	1/49	0/214	0.038
CVD medication (yes/no)	1/49	0/214	0.038
Anticoagulants/thrombocyte-aggregation inhibitors (yes/no)	0/50	0/214	n.s.
Pain medication (yes/no)	0/50	1/213	n.s.
Stimulants (yes/no)	0/50	0/214	n.s.
Other (yes/no)	1/49	0/214	0.038
**Distraction (yes/no)**	18/32	36/177	0.01
Missing	0	1	
Smartphone (yes/no)	2/48	2/211	n.s.
Talking to someone (yes/no)	4/46	4/209	0.023
External factors (yes/no)	3/47	18/195	n.s.
In thoughts (yes/no)	6/44	5/208	**0.002**
Listening to music (yes/no)	1/40	0/213	n.s.
Other (yes/no)	0/50	8/205	n.s.
Missing	6	1	
**Currently stressed (yes/no)**	25/24	62/152	0.003
Missing	6	0	
**Stressed before accident (yes/no)**	18/32	35/179	**0.002**
Missing (yes/no)	6	0	
Argument (yes/no)	5/45	3/211	**0.001**
Pressure situation (yes/no)	10/40	20/194	0.033
Bereavement (yes/no)	0/50	0/214	n.s.
Other (yes/no)	2/48	8/206	n.s.
Missing	6	0	
**Overconfidence (Yes/no)**	19/31	30/183	**0.0004**
**Lack of sleep (yes/no)**	10/40	19/195	0.058
**Previous accidents (yes/no)**	21/28	35/179	**<0.0001**

Differences in age between ADHD and control groups were calculated by student’s test and results given as mean ± standard deviation (SD). Differences in circumstances of the accidents were calculated by χ²-test. Level of significance was set at *p* < 0.0023. Significant *p*-values are in bold. ADHD = attention-deficit/hyperactivity disorder; m = male; f = female; CVD = cardiovascular disease.

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
