# Peer review of "Prevalence of ADHD in Accident Victims: Results of the PRADA Study"

_jcm, 2019, doi:10.3390/jcm8101643_

Round 1

Reviewer 1 Report

The paper  “Prevalence of ADHD in accident victims: results of the PRADA study” investigates the prevalence and potential mechanisms of increased accident risk in adult patients with ADHD. The study covers an interesting area of research, whereby the manuscript has some shortcomings where the authors should take a closer look again.

To me, it remains unclear, which kind of trauma the subjects included in the study experienced. How much time passed between the accident and the assessments?

Therefore, it remains unclear how far this could influence interpretation of results.

In section 3.3 (line 186, line 202, table 2, table 3), the authors should not report “trend level” differences. There is no ‘almost rejected’ of H0 when p-values approximate but are somewhat higher than the pre-set α.

Concerning statistical analyses, some control for multiple testing should be conducted to avoid false positives.

In the discussion, the authors mention a potentially less conservative approach to estimate prevalence of ADHD in the sample investigated. However, I don’t see reason why this approach should be used here as the authors themselves demonstrate that the comprehensive diagnostic process (expectedly) results in fewer cases as the screening suggests.

I strongly recommend the authors to perform spell-check throughout the manuscript and check multiple/wrong spaces, commas, decimal places (within entities) but also English sentence structure and mode of expression.

Table 1: I think there is missing a “.” in the second row of the p-column; same for line 159.

Table 1 and 2: In the legend, level of significance most probably was meant to be p =< 0.005 instead of “p>=”, however, this would not be in line with the earlier reported significance level “p ≤ 0.05” (line 127).

Report of statistical results is not consistent. Spaces around “=” aren’t kept constant; same for “≤” vs “=<”.

Line 208: it most probably was meant to be “p <= 0.05”

Line 217: This should be table 3. In table 2 and table 3, the last column should have the header “p” instead of “P”.

In table 3, information on age and sex are redundant.

Author Response

Response to reviewers

Reviewer 1:

The paper  “Prevalence of ADHD in accident victims: results of the PRADA study” investigates the prevalence and potential mechanisms of increased accident risk in adult patients with ADHD. The study covers an interesting area of research, whereby the manuscript has some shortcomings where the authors should take a closer look again.

To me, it remains unclear, which kind of trauma the subjects included in the study experienced. How much time passed between the accident and the assessments?

Therefore, it remains unclear how far this could influence interpretation of results.

The reviewer is right that indeed we did not assess what kind of injuries the patients had only the circumstances and we did also not report and not assess in any case the time between accident and assessment. However, as we did not recruit patients from neurosurgery units, there were no patient with severe head trauma which could have biased the results. We added this limitation to the discussion section as follows:

“Limitations of our study are that we did not assess injury type or time between trauma and assessment as well as no detailed medical record data.”

In section 3.3 (line 186, line 202, table 2, table 3), the authors should not report “trend level” differences. There is no ‘almost rejected’ of H0 when p-values approximate but are somewhat higher than the pre-set α.

The reviewer has a point about not reporting trend levels, we changed this in the revised manuscript.

Concerning statistical analyses, some control for multiple testing should be conducted to avoid false positives.

The reviewer is right about that and in the revised  manuscript, we have corrected our results for multiple testing and revised the methods as well as the results and discussion section accordingly. Most of the main results however could withstand Bonferroni correction for multiple tests.

“As the accident questionnaire consisted of 22 different items, we applied Bonferroni-correction for multiple testing, and all p-values reported were defined as being significant at the level of p ≤ 0.0023 (0.05/22). Due to the age difference between the ADHD group and the control group, all items which showed significant results in the Chi-square Test/Fisher’s Exact Test were later included in an additional multivariate logistic regression model.

In the discussion, the authors mention a potentially less conservative approach to estimate prevalence of ADHD in the sample investigated. However, I don’t see reason why this approach should be used here as the authors themselves demonstrate that the comprehensive diagnostic process (expectedly) results in fewer cases as the screening suggests.

We thank the reviewer for hinting at this point. It is true that we found less ADHD diagnosis in the full diagnostic assessment compared to the self report. However, we could only conduct the full diagnostic assessment in a minority of the patients and at least in other population, like for example substance abuse patients, there is evidence that using the ASRS self report leads to underreporting of ADHD symptoms rather then overreporting ( Drug Alcohol Depend. 2019 Feb 1;195:52-58. Screening for adult attention-deficit/hyperactivity disorder in alcohol dependent patients: Underreporting of ADHD symptoms in self-report scales. Luderer M et al.). Therefor we additionally reported the less conservative approach using both parts and all items of the ASRS, however, the further analysis were conducted only with the patients who fulfilled the full screening criteria.

Additionally, we explained the assessment and the ASRS screening test itself much more in detail in the revised version of the manuscript, see also answers to Reviewer 2 .

I strongly recommend the authors to perform spell-check throughout the manuscript and check multiple/wrong spaces, commas, decimal places (within entities) but also English sentence structure and mode of expression.

The manuscript was revised by a native speaking co-worker who also helped revising the contents of the manuscript so we added her to the co-author list (Dr. Rhiannon McNeill). We only highlighted the substantial changes in yellow in the revised version of the manuscript but the language was improved in the whole manuscript.

Table 1: I think there is missing a “.” in the second row of the p-column; same for line 159.

Yes, we apologize for this error, we also have revised the whole table 1 in the revised manuscript for a clearer picture of the sample and subgroups along with figure 1.

Table 1 and 2: In the legend, level of significance most probably was meant to be p =< 0.005 instead of “p>=”, however, this would not be in line with the earlier reported significance level “p ≤ 0.05” (line 127).

The reviewer is right, this was a writing error, however, in the revised version of the manuscript, we used the corrected p-value of <0.0023.

Report of statistical results is not consistent. Spaces around “=” aren’t kept constant; same for “≤” vs “=<”.

We corrected this in the revised version of the manuscript.

Line 208: it most probably was meant to be “p <= 0.05”

We corrected this in the revised version of the manuscript, also we report now the p-values corrected for multiple comparison.

10.Line 217: This should be table 3. In table 2 and table 3, the last column should have the header “p” instead of “P”.

We corrected this in the revised version of the manuscript.

In table 3, information on age and sex are redundant.

We changed this in the revised version of the manuscript.

Reviewer 2 Report

Comments to Prevalence of ADHD in accident victims: results of 2 the PRADA study

This manuscript used data from a large clinical sample to investigate the prevalence of ADHD in accident victims and the potential mechanisms of accidents associated with ADHD accident victims. This is a topic with clinical and public health importance, but there are a number of issues that could be further clarified.

1.       P2L79. The patients were identified with ICD diagnosis S00-T98. Please clarify if only primary diagnosis is considered or secondary diagnosis as well. Also, is there any information on the cause of accidents? For example, for poisoning (T36-T50), could you tell if they were accidental or intentional?

2.       P2L82. Please provide the full name for first use of abbreviations (ie. ASRS).

3.       P3L99. Figure 1. A flowchart may better illustrate the sample selection process.

4.       P3L112. Accident Questionnaire. Is there any information from the medical records that may validate (some of) the self-report questions? For example, type of accident or substance use.

5.       P3L123. How is the control group with questionnaire (n=214) selected, and any difference from the rest of controls?

6.       P5L173. “After including age and all the variables that…” Please list the adjusted variables here.

7.       P5L182 “Throughout the whole sample, most accidents happened in road traffic” A table with the type/cause of the accidents would be helpful.

8.       P5L203. It seems a multivariate logistic regression has been conducted. Consider adding a table with the coefficients of all the independent variables and confidence intervals in addition to the p values.

9.       P6L217. It should be “Table 3”.

Author Response

Reviewer 2

This manuscript used data from a large clinical sample to investigate the prevalence of ADHD in accident victims and the potential mechanisms of accidents associated with ADHD accident victims. This is a topic with clinical and public health importance, but there are a number of issues that could be further clarified.

P2L79. The patients were identified with ICD diagnosis S00-T98. Please clarify if only primary diagnosis is considered or secondary diagnosis as well. Also, is there any information on the cause of accidents? For example, for poisoning (T36-T50), could you tell if they were accidental or intentional?

The patients were identified with those diagnoses for recruitment but for the sake of recruiting a great number of patients, we did not extract further medical data from the hospital data system, instead we only ask the patients to give a self report on the circumstances and mechanisms of the accident or injury they have suffered. As the reviewer could kindly see in the accident questionnaire in the supplemental material, we have asked: where the accident happened (road traffic, sport, work, education, public place), the we asked for details what kind of activity the patient was doing while the accident happened, we asked if the accident was self-inflicted and for potential causes of the accident we asked for alcohol, illegal substances, medication, distraction, stress, overconfidence and lack of sleep.

However, the reviewer is right, more details from the medical records would have been also interesting to include in the analysis.

P2L82. Please provide the full name for first use of abbreviations (ie. ASRS).

We have done that in the revised manuscript.

P3L99. Figure 1. A flowchart may better illustrate the sample selection process.

We thank the reviewer for this valuable suggestion and we have changed Figure 1 to a flowchart.

P3L112. Accident Questionnaire. Is there any information from the medical records that may validate (some of) the self-report questions? For example, type of accident or substance use.

The reviewer is right about that, but as we pointed out in 1. We unfortunately did not assess the medical record in more detail.

P3L123. How is the control group with questionnaire (n=214) selected, and any difference from the rest of controls?

The control group with questionnaire were randomly selected and they did not differ in age or sex from the rest of the control group. We revised the Table 1 to make that hopefully clearer. We did not assess other variables in the rest of the control group, because the estimation of ADHD prevalence was the main aim of the study and we therefore minimized the number of variables to collect in order to make it possible to recruit a higher number of patients.

P5L173. “After including age and all the variables that…” Please list the adjusted variables here.

We have done that in the revised manuscript as follows:

“Additionally, multivariate logistic regression analysis including diagnosis and variables that were identified as significantly different between the ADHD and control groups in the single tests (age, substance influence in general, alcohol, illegal drugs, being in-thoughts, being stressed before the accident, having had an argument and overconfidence) revealed that only being in thoughts and being overconfident remained significantly different between ADHD and controls (see Table 4).”

P5L182 “Throughout the whole sample, most accidents happened in road traffic” A table with the type/cause of the accidents would be helpful.

There was no significant difference regarding the type of accidents we assessed with our accident questionnaire between ADHD and control patients, however, we have added this information as supplemental table 1 to the revised manuscript.

P5L203. It seems a multivariate logistic regression has been conducted. Consider adding a table with the coefficients of all the independent variables and confidence intervals in addition to the p values.

Mainly we have calculated Chi-square tests and in the revised version we also added a correction for multiple comparison. However, we still left that multivariate logistic regression with all the significant variables from the Chi-Square tests in the analysis as additional test and we added a Table 4 in the revised version if the manuscript as suggested. See also reviewers point 6.

P6L217. It should be “Table 3”.

Thank you for hinting at our mistake, we have corrected that in the revised version of the manuscript.

Reviewer 3 Report

Prevalence of ADHD in accident victims: results of the PRADA study

This is an interesting investigation into the links between accidents and ADHD diagnosis. There are some concerns regarding the underlying reasons for post-hoc exploratory analyses, which are not well explained. There is a need for clearer language as some sentences are difficult to read bordering on unintelligible. I have made notes below regarding these concerns that should be taken on board before considering this manuscripts publication.

The Adult ADHD RS needs greater explanation. It is unclear why there was a cut-off point at all. It sounds as if you only looked for those who chose “sometimes” in q’s 1-3 and/or “often” in questions 4-6 in only one part of the questionnaire. You also note that In previous studies this was done, however, there is no reference for these studies therefore the evidence cannot be verified. What was Part B of the questionnaire and why was it different?

Please fix the sentence: “There were additional patients that had less than four questions answered indicatively for an ADHD but scored very high in the part B…”

Please be careful with wording “we could find a prevalence of 6.18 % adult ADHD” – you didn’t “find” anything, there was a prevalence of 6.18%.

Insert a comma on line 136 before ‘which’ and make this entire sentence much clearer.

Line 137: It is an interesting finding, but these kind of statements should be reserved for the discussion.

Line 140: I cannot find a description as to what “positively screened” means – positive for ADHD? What is the difference between part A and part B?

Line 149: “fully persistent and partially persistent ADHD patients together” – What is partially persistent ADHD?

Line 167: “Also a pre-existing diagnosis of ADHD was more common 168 found in our ADHD group” – this seems fairly benign. Your ADHD group had ADHD

Table 2: “Personality Disoders” be careful of spelling errors

Line 223: “In the recent years, it became evident that as well childhood and adolescent ADHD as adult ADHD is associated with an increased risk of unintentional injuries and accidents.” A lot of sentences need re-writing to ensure the message is received and understood

Line 267: Reference needed here

Line 305: This needs to be edited out “Funding: Please add: “This research received no external funding” or “This research was funded by NAME OF FUNDER, grant number XXX” and “The APC was funded by XXX”. Check carefully that the details given are accurate and use the standard spelling of funding agency names at https://search.crossref.org/funding, any errors may affect your future funding.”

Author Response

Reviewer 3

Prevalence of ADHD in accident victims: results of the PRADA study

This is an interesting investigation into the links between accidents and ADHD diagnosis. There are some concerns regarding the underlying reasons for post-hoc exploratory analyses, which are not well explained. There is a need for clearer language as some sentences are difficult to read bordering on unintelligible. I have made notes below regarding these concerns that should be taken on board before considering this manuscripts publication.

The Adult ADHD RS needs greater explanation. It is unclear why there was a cut-off point at all. It sounds as if you only looked for those who chose “sometimes” in q’s 1-3 and/or “often” in questions 4-6 in only one part of the questionnaire. You also note that In previous studies this was done, however, there is no reference for these studies therefore the evidence cannot be verified. What was Part B of the questionnaire and why was it different?

We apologize for not describing clear enough the ASRS assessment and the self report questionnaire itself. We have added more information about that in the revised version of the manuscript as follows:

“The ASRSv1.1 consists of an 18-item checklist which assesses adult ADHD symptoms as reported in the DSM-IV-TR. Part A consists of six items, four regarding attention deficit symptoms and two regarding hyperactivity/impulsivity symptoms. Part A of the ASRS-18 item has shown an even greater sensitivity as the 18-item ASRS and can be used as a separate screening tool, also known as the ASRSv1.1 short form. Participants or patients indicate frequency of occurrence of each item with 0 (never), 1 (rarely), 2 (sometimes), 3 (often) or 4 (very often). The short form of the test can be considered positive when four or more answers are above the cutoff value, which equates to four or more questions answered with >=2 (questions 1-3) and/or >=3 (questions 4-6) in part A of the 18-item ASRS [24]. Part B contains 12 additional items that further assess ADHD items. In total, all 18 items represent each 9 DSM-IV symptoms of inattention and impulsivity/hyperactivity. There are different ways of employing the 18 ASRS items. We only used patients for further analysis that screened positive in part A, not patients that screened only positive when evaluating part A and B together, but were not positive for ADHD only in the short form (=Part A) [25]. We assessed additionally the ASRSv1.1 18 item questionnaire instead of the short form only in order to determine whether sub-analysis regarding sub-groups of ADHD (inattentive vs. hyperactive subtype) could provide further information.”

Please fix the sentence: “There were additional patients that had less than four questions answered indicatively for an ADHD but scored very high in the part B…”

We apologize for the language, the whole manuscript has been revised by a native-speaker (Dr. Rhiannon McNeill), we highlighted only the substantial changes and changes in content, but the language in the whole manuscript was improved by Dr. McNeill, therefore we added her as a co-author. She also contributed to the content modification of the revised manuscript.

Please be careful with wording “we could find a prevalence of 6.18 % adult ADHD” – you didn’t “find” anything, there was a prevalence of 6.18%.

We thank for hinting this out and we changed that in the revised manuscript.

Insert a comma on line 136 before ‘which’ and make this entire sentence much clearer.

The sentence was revised in the manuscript as follows:

Using only the positively screened participants in part A of the ASRS, there was a prevalence of 6.18 % adult ADHD in our sample of accident victims. A less conservative approach would have demonstrated a prevalence of 8.3 %, including the additional participants who screened positively for ADHD only if Parts A and B of the ASRS were evaluated together.”

Line 137: It is an interesting finding, but these kind of statements should be reserved for the discussion.

The sentence has been deleted in the revised version if the manuscript.

Line 140: I cannot find a description as to what “positively screened” means – positive for ADHD? What is the difference between part A and part B?

Again, we apologize for not having explained the screening instrument in more detail, we did this in the revised manuscript and also modified this sentence as follows:

“For further analyses, we used only the group of patients who screened positive for ADHD in Part A (the short version of the ASRS). From herein they are referred to as the ADHD group, and the control group consists of accident victims who screened negative in the ASRS short version.

Line 149: “fully persistent and partially persistent ADHD patients together” – What is partially persistent ADHD?

With this we refer to the patients we mentioned above in the manuscript, having been diagnosed as children with ADHD but not fulfill the full adulthood criteria, that means less than 5 hyperactive/impulsive symptoms and less than 5 inattentive symptoms (but more than 3 each) in adulthood regarding the DSM-5 ADHD diagnostic criteria but fulfilling more than 5 each in childhood (retrospectively questioned by our specialized outpatient clinic).

Line 167: “Also a pre-existing diagnosis of ADHD was more common 168 found in our ADHD group” – this seems fairly benign. Your ADHD group had ADHD

Yes, but this validated the ASRS screening once more. We hope we have clarified this in the revised version of the manuscript as follows:

“A pre-existing, previously confirmed ADHD diagnosis was also significantly more common in our ADHD group (χ²-test= 21.50, p<0.0001) compared to controls, further validating the screening instrument used.”

Table 2: “Personality Disoders” be careful of spelling errors

Corrected.

Line 223: “In the recent years, it became evident that as well childhood and adolescent ADHD as adult ADHD is associated with an increased risk of unintentional injuries and accidents.” A lot of sentences need re-writing to ensure the message is received and understood

As stated above, we apologize and the manuscript was revised by a colleague and native speaker who also edited the content and was therefore added as a co-author.

Line 267: Reference needed here

Reference was added.

Line 305: This needs to be edited out “Funding: Please add: “This research received no external funding” or “This research was funded by NAME OF FUNDER, grant number XXX” and “The APC was funded by XXX”. Check carefully that the details given are accurate and use the standard spelling of funding agency names at https://search.crossref.org/funding, any errors may affect your future funding.”

We edited the funding section according also to the funding sources as follows:

“Funding: This research was funded by Medice Arzneimittel Pütter GmbH & Co KG, Germany, grant PRADA study. This publication was funded by the German Research Foundation (DFG) and the University of Wuerzburg in the funding programme Open Access Publishing.”

Round 2

Reviewer 3 Report

The authors have done well to improve the manuscript. It is not necessary for authors to apologise to reviewers, we are happy to review your work and simply wish to assist in its publication.

I still have some suggestions before the manuscript could be considered for publication that require attention.

I thank the authors for having an additional person review the language and update the MS. There are, however, a number of sentences that require improvement to ensure you are getting across your data.

Regarding the description of the ADHD scale used, there still needs to be greater clarification. Please find below my suggestions:

Use the acronym ASRS-SF (ASRS-Short Form) when reporting results from the Part A only. When discussing the full version, I suggest using the acronym ASRS-18. It will streamline the methods and results sections for greater clarification.

The following is an edited paragraph of the ASRS description. It is entirely up to the authors if they wish to use it. If part of the description is incorrect, then it can be adjusted:

"The ASRSv1.1 is an 18-item checklist that assesses symptoms of ADHD in adult populations. The ASRS contains two components, Part A and Part B. Part A consists of six items; four regarding attention deficit symptoms and two regarding hyperactivity/impulsivity symptoms. Part A of the ASRS has shown equal sensitivity to the full version and is commonly used as a separate screening tool, known as the ASRSv1.1 short form (ASRS-SF). Participants completed the full-version of the ASRS (ASRS-18) indicating the frequency of symptom occurrence 0 (never), 1 (rarely), 2 (sometimes), 3 (often) or 4 (very often). The ASRS-SF can be considered positive when four or more answers are above the cut-off value, which equates to four or more questions answered with >=2 (questions 1-3) and/or >=3 (questions 4-6) [24]. Part B consists of 12 additional ADHD symptom assessments. In the current study, patients were only included in further analyses following a positive screen on the ASRS-SF. Those who did not screen positive in the ASRS-SF were not included in primary analyses [25]. Post-hoc analyses were run on the full ASRS-18, regardless of positive screens, to determine any ADHD sub-type influence."

Author Response

Response to Reviewer 3

The authors have done well to improve the manuscript. It is not necessary for authors to apologise to reviewers, we are happy to review your work and simply wish to assist in its publication.I still have some suggestions before the manuscript could be considered for publication that require attention.I thank the authors for having an additional person review the language and update the MS. There are, however, a number of sentences that require improvement to ensure you are getting across your data.Regarding the description of the ADHD scale used, there still needs to be greater clarification. Please find below my suggestions:

Use the acronym ASRS-SF (ASRS-Short Form) when reporting results from the Part A only. When discussing the full version, I suggest using the acronym ASRS-18. It will streamline the methods and results sections for greater clarification.

Thank you for this suggestion, we changed it in the revised version of the manuscript.

The following is an edited paragraph of the ASRS description. It is entirely up to the authors if they wish to use it. If part of the description is incorrect, then it can be adjusted:

"The ASRSv1.1 is an 18-item checklist that assesses symptoms of ADHD in adult populations. The ASRS contains two components, Part A and Part B. Part A consists of six items; four regarding attention deficit symptoms and two regarding hyperactivity/impulsivity symptoms. Part A of the ASRS has shown equal sensitivity to the full version and is commonly used as a separate screening tool, known as the ASRSv1.1 short form (ASRS-SF). Participants completed the full-version of the ASRS (ASRS-18) indicating the frequency of symptom occurrence 0 (never), 1 (rarely), 2 (sometimes), 3 (often) or 4 (very often). The ASRS-SF can be considered positive when four or more answers are above the cut-off value, which equates to four or more questions answered with >=2 (questions 1-3) and/or >=3 (questions 4-6) [24]. Part B consists of 12 additional ADHD symptom assessments. In the current study, patients were only included in further analyses following a positive screen on the ASRS-SF. Those who did not screen positive in the ASRS-SF were not included in primary analyses [25]. Post-hoc analyses were run on the full ASRS-18, regardless of positive screens, to determine any ADHD sub-type influence."

 Thank you very much, the content is correct so we inserted it in the revised manuscript.

This manuscript is a resubmission of an earlier submission. The following is a list of the peer review reports and author responses from that submission.